# The Photocatalytic Conversion of Carbon Dioxide to Fuels Using Titanium Dioxide Nanosheets/Graphene Oxide Heterostructure as Photocatalyst

**DOI:** 10.3390/nano13020320

**Published:** 2023-01-12

**Authors:** Apisit Karawek, Kittipad Kittipoom, Labhassiree Tansuthepverawongse, Nutkamol Kitjanukit, Wannisa Neamsung, Napat Lertthanaphol, Prowpatchara Chanthara, Sakhon Ratchahat, Poomiwat Phadungbut, Pattaraporn Kim-Lohsoontorn, Sira Srinives

**Affiliations:** 1Nanocomposite Engineering Laboratory (NanoCEN), Department of Chemical Engineering, Faculty of Engineering, Mahidol University, Nakhon Pathom 73170, Thailand; 2Department of Chemical Engineering, Faculty of Engineering, Mahidol University, Nakhon Pathom 73170, Thailand; 3Center of Excellence on Catalysis and Catalytic Reaction Engineering, Department of Chemical Engineering, Faculty of Engineering, Chulalongkorn University, Bangkok 10330, Thailand

**Keywords:** titanium dioxide, CO_2_ conversion, photoreduction, photocatalyst, graphene

## Abstract

Carbon dioxide (CO_2_) photoreduction to high-value products is a technique for dealing with CO_2_ emissions. The method involves the molecular transformation of CO_2_ to hydrocarbon and alcohol-type chemicals, such as methane and methanol, relying on a photocatalyst, such as titanium dioxide (TiO_2_). In this research, TiO_2_ nanosheets (TNS) were synthesized using a hydrothermal technique in the presence of a hydrofluoric acid (HF) soft template. The nanosheets were further composited with graphene oxide and doped with copper oxide in the hydrothermal process to create the copper−TiO_2_ nanosheets/graphene oxide (CTNSG). The CTNSG exhibited outstanding photoactivity in converting CO_2_ gas to methane and acetone. The production rate for methane and acetone was 12.09 and 0.75 µmol h^−1^ g_cat_^−1^ at 100% relative humidity, providing a total carbon consumption of 71.70 µmol g_cat_^−1^. The photoactivity of CTNSG was attributed to the heterostructure interior of the two two−dimensional nanostructures, the copper−TiO_2_ nanosheets and graphene oxide. The nanosheets−graphene oxide interfaces served as the n−p heterojunctions in holding active radicals for subsequent reactions. The heterostructure also directed the charge transfer, which promoted electron−hole separation in the photocatalyst.

## 1. Introduction

Carbon dioxide (CO_2_) emissions into the atmosphere is an issue that has garnered significant attention worldwide. Emissions have contributed to global warming and climate change. They are a cause of respiratory-related problems in humans and animals [1,2]. Following the Paris agreement at the 2015 United Nations Climate Change Conference, the 196 signatory nations had the ambitious goal of limiting the average global temperature rise to 2 °C above the pre-industrial level [3]. Half of all global CO_2_ emissions need to be cut by 2030. The point of net zero emissions must be reached by 2050 to achieve the goal. Current technologies concerning carbon capture and storage (CCS) [1,2,4] involve capturing CO_2_ with an absorbent/adsorbent or storing CO_2_ underground. CO_2_ capture requires the absorbent/adsorbent to be regenerated and reused, consuming energy and eventually emitting more CO_2_ into the atmosphere. CO_2_ storage in an underground space is at risk of leakage and is burdened with long-term inspections [2,4,5]. CO_2_ conversion to a high–value product is an alternative to CCS. CO_2_ is reformed into other substances with assistance from a photocatalyst, such as zinc oxide (ZnO) [6,7], cadmium sulfide (CdS) [8] or titanium dioxide (TiO_2_) [2,9,10]. The substances can be used in industrial plants or sold as commodities.

TiO_2_ is a popular photocatalyst used in the catalytic conversion of CO_2_ gas to products such as methane (CH_4_) [9,11], methanol [9,10,12] and ethanol [9,10,12]. The reaction involves water dissociation (Equation (1)), which results in electron transfer to the TiO_2_ valence band (VB). The VB electrons are boosted to the conductive band (CB) and become photoinduced (Equation (2)). The photoelectrons can incorporate with CO_2_ (Equation (3)), yielding the CO_2_^−^ radical, which serves as an intermediate for subsequent reactions, producing products (Equations (4)–(9)) [10,11,12].
H_2_O → 2H^+^ 1/2O_2_ + 2e^−^ (VB)(1)
2e^−^ (VB) → 2e^−^ (CB)(2)
2CO_2_ + 2e^−^(CB) → 2CO_2_^−^(3)
2CO_2_^−^ + 14e^−^ (CB) + 16H^+^ → 2CH_4_ + 4H_2_O(4)

CO_2_ photoreduction occurs in the gas and liquid phases. The gas–phase operation is relatively slow and yields small hydrocarbon products since it deals with diluted analyte concentrations and competitive adsorption on photocatalysts. [13,14]. Olivo and his team [15] investigated the reaction mechanisms and highlighted the vital role of water, which adsorbs on the photocatalyst and reacts with CO_2_^−^ through a hydrogenation reaction (Equation (5)). The reaction produces formic acid (HCOOC), which serves as an intermediate for the formation of more extensive hydrocarbon products, such as acetic acid (Equation (6)) and methanol (CH_3_OH) (Equation (7)). In case the water was not sufficient, the CO_2_^−^ underwent a different path of deoxygenation (Equation (8)), freeing active carbon atoms for the formation of methane (Equation (9)).
CO_2_^−^ + 2H^+^ + e^−^ → HCOOH(5)
HCOOH + 2H^+^ 2e^−^ → HCHO + H_2_O(6)
HCHO + 2H^+^ + 2e^−^ → CH_3_OH (7)
CO_2_^−^ → C + O_2_ + e^−^(8)
C + 4H^+^ + 4e^−^ → CH_4_
(9)

The photoactivity of TiO_2_ is limited by wide bandgap energy and the quick pairing rate between the photoinduced electrons and holes, which leads to insufficient photoactivity [2,16]. Three approaches could be applied to enhance the photoactivity of TiO_2_: synthesis of nanostructures, the addition of a metal co–catalyst, and compositing TiO_2_ with carbon nanostructures. TiO_2_ nanostructures can be synthesized using various techniques, such as sol–gel [17], hydrothermal [10], sonochemical [18], hard template [19] and soft-template syntheses [12,20,21]. A hard template synthesis involves using rigid porous materials that control the growth of the nanostructures, while a soft template synthesis utilizes chemicals that provide intermolecular interactions with a precursor. Various soft templates, such as sodium hydroxide (NaOH), potassium hydroxide (KOH) and hydrofluoric acid (HF), were demonstrated for the synthesis of TiO_2_ nanorods [22], nanotubes [23], nanoribbon [12] and nanosheets [20,21]. The nanostructures provide high surface activity and a high surface–to–volume ratio, which amplifies the photoactivity of TiO_2_. Adding metal and metal oxide as a co-catalyst to TiO_2_ yields a transition state for the photoelectrons to be separated from holes [10,11,17,18]. The separation reduces the recombination rate for electrons and holes and allows more photoelectrons to facilitate a photoreaction. Metals such as silver [10], palladium [24] and copper [9,10,12,24] have been demonstrated to display co–catalyst ability. Carbon nanostructures, such as carbon nanotubes and graphene, contain structural defects that draw charges and radicals. The defects offer sites for TiO_2_ precursors to accumulate and deposit, thereby becoming TiO_2_/carbon composites [12,13,25]. Interfaces between TiO_2_ (n–type semiconductor) and carbon (p–type semiconductor) behave as an n–p heterojunction, in which charges and radicals are accommodated and stabilized [13,25]. The radicals are later involved in accelerating the photoreaction.

Heterostructures of two 2–dimensional nanostructures (2D–2D heterostructure) were demonstrated to be excellent interiors for photocatalysts [12,13,25]. The outstanding photoactivity of the 2D–2D heterostructure results from a charge separation ability, a directed path for charge mobility, and enhanced light adsorption. Lertthanaphol [10] synthesized a copper–doped sodium dititanate nanosheets/graphene oxide (GO) heterostructure using a one-step hydrothermal technique with assistance from a sodium hydroxide soft template. The heterostructure converted dissolved CO_2_ to liquid fuels, such as methanol and ethanol, at a remarkable production rate in hundreds of µmol g_cat_^−1^h^−1^. Tan and his team [14] composited oxygen–rich TiO_2_ nanoparticles with GO through a wet impregnation technique. The TiO_2_/GO composite photo–reduced CO_2_ gas in the presence of water vapor, yielding gas products of methane (4 µmol g_cat_^−1^h^−1^), carbon monoxide (CO) (15 µmol g_cat_^−1^h^−1^), ethane (0.4 µmol g_cat_^−1^h^−1^) and ethylene (4 µmol g_cat_^−1^h^−1^). Xiong and his team [26] synthesized TiO_2_ nanosheets on GO by adding titanium butoxide in an ethanol medium where GO was suspended. The HF was added to the mixture as a soft template in the hydrothermal process to form TiO_2_ nanosheets. The TiO_2_ nanosheets are 10–30 nm in size and exhibited anatase crystallography with extra exposure of {001} facets. Photoactivity of the TiO_2_ nanosheets/GO was revealed in a gas–phase CO_2_ photoreduction, in which methane (25 µmol g_cat_^−1^h^−1^) and CO (70 µmol g_cat_^−1^h^−1^) were obtained.

Graphene is a 2D carbon nanostructure with excellent charge transfer, chemical stability, and unique electronic properties [10,12,13,14]. It has been used for various applications, including additives for mechanical reinforcement, coating for abrasive resistance, cathodes in batteries and catalyst support. Chemical exfoliation of graphite yields a gram scale of graphene in the form of graphene oxide (GO), which contains chemical functionalities, including carboxyl, carbonyl and hydroxyl. The functionalities induce cation accumulation and precipitation, which assists in forming and immobilizing metal/metal oxide nanostructures on GO. The TiO_2_/graphene composites have been demonstrated as highly active catalysts due to the nature of n–p heterojunctions and the high surface photoactivity of the TiO_2_.

In this work, we aimed to present a heterostructure of copper–doped TiO_2_ nanosheets/GO composite (CTNSG) using a one–step hydrothermal technique in the presence of HF. The CTNSG was used as a catalyst in the photoreduction of humid CO_2_ gas to gas and vapor products. The composites were analyzed for chemical properties, crystallography, optical properties, and physical geometry to understand the characteristics of the photocatalyst.

## 2. Materials and Methods

All chemicals were of analytical grade and used with no further treatment. Graphite flakes (Alfa Aesar, Haverhill, MA, USA 99.9%, 325 mesh), sodium nitrate (Fluka chemika, Buchs, Switzerland 99% NaNO_3_), potassium permanganate (Ajax FineChem, New South Wales, Australia 99.0% KMnO_4_), sodium hydroxide (Ajax FineChem, NaOH), copper (II) nitrate (Sigma Aldrich, St. Louis, MI, USA Cu(NO_3_)_2_⋅3H_2_O), and sodium borohydride (Sigma Aldrich, NaBH_3_) were purchased and used as received. Sulfuric acid (RCI Labscan, Bangkok, Thailand 98% H_2_SO_4_), hydrochloric acid (RCI Labscan, 37% HCl), HF (Qrec, Chon Buri Chonburi, Thailand, 49%), hydrogen peroxide (Merck, Rahway, NJ, USA, 30% H_2_O_2_), ethanol (RCI Labscan, 99.9% C_2_H_5_OH), and titanium (IV) butoxide (Sigma Aldrich, Ti(OCH_2_CH_2_CH_2_CH_3_)_4_) were used as received. Iso–Propanol (IPA, (CH_3_)_2_CHOH) was purchased from RCI Labscan and used without further treatment. The CO_2_ gas (99.9% purity) was purchased from Lor Ching Tong Oxygen (Samut Sakhon, Thailand).

### 2.1. GO Synthesis

GO was synthesized through a chemical exfoliation process in a fume hood. The graphite flakes of 2 g were mixed with 1 g of sodium nitrate in 50 mL of concentrated sulfuric acid. The mixture was stirred while being chilled in an ice bath at 0 °C for 2 h. During this time, 7.3 g potassium permanganate was poured gradually into the mix while the temperature was maintained below 4 °C. The mixture was transferred and stirred at ambient conditions for 2 h, allowing the graphite to be oxidized and become graphene oxide (GO). The oxidation reaction was terminated by adding 55 mL DI water and 7 mL hydrogen peroxide as the excess manganese ions were converted to acid–soluble manganese oxide. The mixture was filtered using a vacuum filtration apparatus to obtain a brown GO slurry. This slurry was rinsed sequentially with 3% (*v/v*) HCl solution and DI water. The GO was dried in a vacuum oven at 60 °C for 24 h. GO powder was sonicated and resuspended in DI water for powder purification. The suspension was centrifuged at 10,000 rpm for 20 min. The filtrate was discarded to remove manganese and acid residuals from GO. Fresh water was added to resuspend the precipitated GO for another centrifuge round. The cycle continued until the pH 7 filtrate was achieved. The GO was dried in a vacuum oven at 60 °C for 24 h, ground using an analytical-grade pestle and mortar, and kept in a desiccator for future use.

### 2.2. Synthesis of TiO_2_ Nanosheet (TNS) and TiO_2_ Nanoparticle (TNP)

For the synthesis of TNS, 5 mL titanium butoxide was dissolved in a mixture of 10 mL IPA and DI water medium mixture (10:1, *v/v*) while stirred. HF (49%wt) of 0.6 mL was slowly added to the stirred mixture, yielding a visible cloud of titanate gels. The mixture was transferred to a Teflon–lined autoclave reactor for a hydrothermal treatment at 180 °C for 16 h. The process yielded the TNS, obtained via centrifugation. It was rinsed 3 times sequentially with DI water and ethanol and dried in a vacuum oven at 80 °C. TNP was prepared without HF and used as a control sample.

The presence of fluoride ions on TNS affected the photoactivity of the catalyst [27,28], which led us to a study on post–synthesis treatment. Two treatment methods were tested: a thermal treatment and an alkaline treatment. The thermal treatment involved heating the TNS at 600 °C for 90 min (TNS–Thermal). The alkaline treatment concerned suspending the TNS in 0.1 M NaOH while stirring for 3 h, followed by sequential rinsing with DI water and ethanol (TNS–Alkaline). The TNS was dried in a vacuum oven at 80 °C and kept inside a desiccant for future use.

### 2.3. Synthesis of the TNS/GO Composites

The TNS was further used for synthesizing the TNS/GO composites. The TNS of 200 mg was suspended in 10 mL ethanol in a water solution (30% *v/v*) and 3.8 mg Cu(NO_3_)_2_·3H_2_O and 10 mL of GO suspension (0.2 mg mL^−1^) were added. The mixture was stirred for 30 min and was sonicated at 300 Watts for 1 h. Then, 25 mg NaBH_3_ was added, and the mixture was sonicated at 300 Watts for another 1 h. We transferred the mix to the autoclave reactor and held the mix at 180 °C for 6 h. Copper-doped TNS/GO composite (CTNSG) precipitated powder was obtained and rinsed sequentially with ethanol and water. The powder was dried at 80 °C for 12 h and kept in a desiccator. Control samples of TNS/GO (TNSG), TNP/GO (TNPG) and copper–doped TNPG (CTNPG) were also synthesized following the same procedures. In brief, the TNSG was synthesized by suspending TNS in an ethanol solution, and the GO suspension was added to the mix. The mixture was thermally treated in the autoclave reactor to obtain TNSG powder. For TNPG, the TNP in ethanol solution was mixed with the GO suspension and heated in the autoclave reactor. The CTNPG was obtained following similar procedures as the CTNSG replacing TNS with TNP.

### 2.4. CO_2_ Photoreduction

The photoreduction experiments were conducted in a 60 mL gas–tight photoreactor with a quartz window (19.6 cm^2^) for illumination. The mass–controlled CO_2_–feed line purged CO_2_ gas through a water–containing bubbler, introducing a water–saturated CO_2_ stream to the photoreactor dome. The photocatalyst was coated as a thin film on one side of a 1.5 × 1.5 cm^2^ glass slide and employed in the photoreactor dome to facilitate CO_2_ photoreduction. A suspension of photocatalyst powder was prepared in DI water at a concentration of 10 mg mL^−1^, then slowly dropped on the glass slide and heated at 105 °C for 30 min. A total suspension volume of 1 mL was coated on the slide to realize a uniform thin film. In operation, the water-saturated CO_2_ stream filled the glass dome at 0.5 L per min for 20 min and was sealed inside the photoreactor. The photocatalyst inside the photoreactor was illuminated by a mercury lamp (Philips; 160 Watts) for 5 h before a gas sample was collected (1 mL) using a gas–tight syringe (1 mL Manual Syringe, PTFE–tip Plunger, Agilent Technologies). The gas sample was analyzed using gas chromatography (GC PerkinElmer Clarus 680) with a flame ionization detector (FID) and a DB–Wax column for composition analysis. The analysis included preheating the column at 45 °C for 3 min, warming it to 70 °C for 2.5 min, and holding it at 200 °C. The injector and detector were held at 200 °C during the operation.

### 2.5. Sample Characterizations

The JSM 7610F PLUS field emission scanning electron microscope (FE–SEM, JEOL Ltd., Tokyo, Japan) and HR–TEM (Tecnai 20, Philips, Field Electron and Ion Company, Hillsboro, OR, USA) were used for a physical geometry analysis. The D2 Phaser diffractometer X–ray diffraction (XRD, Bruker, Germany) was used to analyze the crystal structures of the sample materials. The XRD relied on a Cu Kα radiation source and was operated at 2θ scanning range of 5 to 80° and a scanning rate of 0.02° s^−1^. The Kratos Axis Ultra DLD X–ray photoelectron spectroscopy (XPS, Kratos Analytical Ltd., Manchester, UK) was used to analyze the atomic composition and bonding states of samples. The JASCO FT/IR–6800 Fourier transform infrared spectrometer (FT–IR, Hachioji, Tokyo, Japan)) characterized IR responses in the 400–4000 cm^−1^ range. The IR responses were cross–analyzed using the HORIBA XploRA PLUS confocal Raman microscope (Horiba, Lille, France). Brunauer–Emmett–Teller (BET) surface areas and porosities were analyzed using an automated gas sorption analyzer (Autosorp iQ–MP–MP, Quantachrome Instruments, Boynton Beach, FL, USA). Before the adsorption measurement, a powder sample was degassed at 150 °C under a high vacuum for 12 h to remove water and other liquid residues. The optical properties and bandgap energies were analyzed by plotting light absorbances from the SHIMADZU UV–1800 spectrophotometer (Shimadzu, Kyoto, Japan) following Tauc’s correlation. The JASCO FP–6200 Photoluminescence (Jasco, Tokyo, Japan) was utilized in analyzing the photoactivity of the photocatalyst.

## 3. Results and Discussion

### 3.1. Material Characterizations

The physical geometry of TNS, TNP and composites was characterized using the FE–SEM (Figure 1). We observed the effects of medium composition on TNS synthesis. The TNS (Figure 1a) and TNP (Figure 1b) were synthesized first in the presence, then in the absence of the HF soft template. The TNS was 20.3 ± 0.4 nm in size with a thickness of 4–5 nm. In comparison, the TNP was an irregular–shaped structure with lumps of particles having an average diameter of 320.6 ± 5.5 nm. The roles of the HF soft template agreed well with the report from Han and his team [20,21] that the fluorine ions promoted the lateral growth of the {001} over the {101} titanate planes. The {001} planes sandwiched titanate structures led to the formation of TNS. The role of the IPA was to enhance the effects of the HF soft template in creating TNS, but it was not a soft template on its own. Synthesis of the TNS with no addition of DI water (HF+IPA) produced TNS with 31.6 ± 0.8 nm width and 4.5 ± 0.1 nm thickness (Figure 1c), giving a larger sheet than, and equal thickness to the original synthesis conditions. In the absence of IPA (DI + HF, Figure 1d), the TNS was 25.81 ± 0.6 nm in width and 7.8 ± 0.2 nm in thickness.

The post–synthesis treatment was utilized to remove fluoride ions (F), which could potentially affect the physical geometry of the TNS. The thermal and alkaline treatments yielded TNS–Thermal (Figure 1e) and TNS–Alkaline (Figure 1f). The alkaline treatment showed no significant effect on TNS geometry (21.2 ± 0.2 nm). In contrast, the thermal treatment transformed TNS into nanoparticles with an average size of 17.8 ± 0.6 nm. The alkaline treatment was then applied as a post–synthesis process for fluoride ion removal. TNSG (Figure 1g) and CTNSG (Figure 1h) displayed TNS or CTNS structures decorated on GO sheets. The TNSG was 23.0 ± 0.5 nm in width and 6.6 ± 0.1 nm thick. The CTNSG was larger than TNSG, with 29.7 ± 0.4 nm in width and 6.7 ± 0.1 nm thick due to the thickness of copper deposition. TNPG (Figure 1i) and CTNPG (Figure 1j) exhibited a synergic effect between TNP and GO as the TNP disintegrated into small nanoparticles and immobilized on GO sheets. The average size of TNP and CTNP on TNPG and CTNPG was 11.6 ± 0.5 nm and 13.2 ± 0.5 nm, respectively. Collective data on the dimensions of the photocatalysts can be found in Appendix A.

We studied atomic composition at the surfaces of CTNSG and CTNPG. SEM–EDS elemental analysis (Appendix A) displayed a homogeneous distribution of copper (Cu), Ti and O on the TNS in CTNSG (Appendix A) and the TNP in CTNPG (Appendix A). The quantitative EDX analysis (Appendix A) showed that the as–synthesized TNS, TNS-Alkaline and TNS-Thermal were composed of carbon (C) and oxygen (O). TNSG and TNPG exhibited a significant amount of carbon, while CTNSG and CTNPG presented a small weight fraction of Cu. The CTNSG and CTNPG were also analyzed by TEM (Figure 2). CTNSG is a 2D–2D heterostructure with face–to–face contacts between the 2D nanostructures of TNS and GO sheet (Figure 2a). CTNPG presents a 0D–2D heterostructure with point–to–face interfaces between the 0D–nanostructure TNP and the 2D nanostructure GO (Figure 2b). In both cases, the lattice fringes were determined to be 0.35 nm (Figure 2a (Inset) and Figure 2b (Inset)), which correlated to the distance between the {101} facets of anatase TiO_2_.

The GO, TNS, TNP and composite crystal structures were analyzed using XRD (Figure 3). For GO, the XRD spectrum reveals a sharp peak at 13.8°, correlating to the {001} graphitic oxide lattice plane [25]. The other broad peak at 23° is ascribed to the {002} plane of the reduced graphene oxide (rGO), indicating a partial reduction of GO to rGO [29]. Both the TNS and TNP exhibit XRD peaks at 25.50°, 38.10°, 48.30°, 54.10°, 55.36°, 62.80°, 68.91°, 70.45°, and 75.24°, which are indexed to the {101}, {004}, {200}, {105}, {211}, {204}, {116}, {220} and {215} planes of the anatase TiO_2_. Anatase is the most photoactive phase of TiO_2_ [2,21], providing excellent light absorbance, a large surface area, and a reasonable electron–hole recombination rate [21,25]. For TNSG, TNPG, CTNSG and CTNPG, the patterns resemble those of the TNS and TNP, in which all the peaks correlate to the anatase TiO_2_. We observed no XRD signal from GO, copper, copper oxides or other impurities on TNS, TNP and the composites. This result can be explained since copper and copper oxide exist only in trace amounts, and GO is not the main contributor to the crystallography of the composites. We determined the crystallite size following Scherrer’s equation (Equation (10)),
D_p_ = kλ(βCosθ)(10)
in which D_p_: crystallite size (nm), k: Scherrer constant (0.9 for TNS and 0.94 for TNP), λ: X–ray wavelength (1.54178 °A for Cu Kα), β: FWHM (Full Width at Half Maximum) of XRD peak, θ: XRD peak position (2θ/2). The crystallite size is an average X–ray diffraction unit inside the material and correlates to the grain size of the crystal structure. The bigger the crystallite size, the more crystalline the structure becomes. GO, TNS, TNP, TNSG, TNPG, CTNSG and CTNPG exhibit crystallite sizes of 4.3, 15, 9.9, 14.0, 9.4, 14.5 and 9.9 nm, respectively. We can explain this trend since GO is relatively less crystalline than TiO_2_, and the TNS provides a bigger grain size than the TNP. Compositing TNS (15 nm) or TNP (9.9 nm) to the GO (43 nm) reduces the total crystallinity of the composite material, resulting in smaller crystallite size for the TNSG (14.0 nm) and TNPG (9.4 nm). The addition of copper to the composites promotes the crystallinity of the materials, as observed in the case of CTNSG (14.5 nm) and CTNPG (9.9 nm).

XPS was utilized to investigate the chemical composition of the composites (Figure 4). For the CTNSG (Figure 4a), a wide scan reveals characteristic peaks at binding energies of 918, 527, 456, and 283 eV, corresponding to Cu 2p, O 1s, Ti 2p, and C 1s, respectively. The C 1s narrow scan (Figure 4b) peaks at binding energies of 289, 288, and 286 eV, indicating the presence of O–C=O, C=O, and C–C/C=C/C–H. The binding energy of 284 eV is correlating to TNS–GO interactions through the C–Ti bond [10,12,13]. The Ti 2p narrow scan (Figure 4c) peaks at 464 and 458 eV, correlating to Ti 2p1/2 and Ti 2p3/2 and representing the Ti^4+^ group of the bulk TiO_2_ [10,13]. For the CTNPG, a wide scan (Figure 4d) shows binding energies at 917, 527, 456, and 283 eV, corresponding to Cu 2p, O 1s, Ti 2p, and C 1s components. The C 1s narrow scan (Figure 4e) exhibits the presence of O–C═O (290 eV), C=O (288 eV), C–C/C=C/C–H (286 eV), and C–Ti groups (284 eV). The Ti 2p narrow scan (Figure 4f) presents Ti 2p1/2 and Ti 2p3/2 bands at binding energies of 463 and 457 eV. It is worth noting that the C–Ti peak is more significant for the CTNSG narrow scan than the CTNPG. This result supports the idea that the TNS–GO interaction of CTNSG is more vital than the TNP–GO of CTNPG.

Raman spectra for GO, TNS, TNP, TNSG, TNPG, CTNSG and CTNPG are presented in Figure 5a. For GO, the spectrum exhibits typical D and G bands at 1350 and 1597 cm^−1^, which can be ascribed to the қ-point phonon of the A_1g_ symmetry and E_2g_ phonon of the sp^2^ carbon [12,26]. The I_D_/I_G_ ratio of defective and crystalline graphitic structures is determined to be 0.91. TNS and TNP present characteristic peaks of anatase TiO_2_ at 145, 196, 399, 518, and 639 cm^−1^ with no detectable signals from rutile and brookite. The Raman peaks located at 639 cm^−1^ (E_g_) and 518 cm^−1^ (A_1g_ + B_1g_) correlate to the Ti–O stretching, while the peak at 399 cm^−1^ (E_g_) is regarded as O–Ti–O bending. For TNSG and TNPG, Raman bands provide the signals of anatase TiO_2_. The D and G bands from the graphene structure are observed with the I_D_/I_G_ ratio of 0.85 (TNSG) and 0.96 (TNPG). CTNSG and CTNPG reveal the Raman bands for the anatase TiO_2_ and D and G bands of GO. The I_D_/I_G_ of 0.88 and 0.73 are determined for CTNSG and CTNPG, respectively.

The FT–IR analysis provides information on the chemical functionalities of the solid samples (Figure 5b). For GO, the FTIR spectrum reveals transmittance peaks at 3300, 1730, 1570, 1230 and 930 cm^−1^, which correspond to −OH stretching, C=O stretching, C=C aromatic, C–O stretching, and C–H bending, respectively [10,12]. Functional groups on GO structure facilitate in attracting metal cations, leading to the formation of metal nanoparticles. The particles are formed and immobilized on GO, creating metal oxide–GO interfaces that stabilize active radicals for subsequent reactions. The pure TiO_2_ sample, including TNS and TNP, displays much less functionality than the GO and GO composite samples. The only detected peak is a broad peak starting from 900 to 400 cm^−1^, representing the bending vibration of Ti–O and Ti–O–Ti. The TNSG and TNPG combine functionalities of the TiO_2_ and GO, yielding transmittance peaks at 3300 cm^−1^ (OH stretching) and 1570 cm^−1^ (C=C aromatic) and a broad peak at 900 to 400 cm^−1^ (Ti–O–C and Ti–O–Ti). For CTNSG and CTNPG, the combination of TiO_2_ and GO functionalities are observed with no detectable signal from the copper. FT–IR transmittance from CTNSG and CTNPG peaks at 3300, 1570, and 900 to 400 cm^−1^ range, which correlates to OH stretching, C=C aromatic, and Ti–O–C and Ti–O–Ti, respectively [12,14].

The automated gas adsorption analyzer provided the surface area and porosities for the GO, TNS, TNP and composites (Appendix A). GO displayed the BET surface of 50.2 m^2^ g^−1^ with the BJH desorption pore of 3.9 nm [30]. TNS and TNP revealed BET surface and mesopore size of 110.4 m^2^ g^−1^ and 10.1 nm and 138.2 m^2^ g^−1^ and 13.2 nm, respectively. TNS yielded a significantly higher surface area and bigger pore size than GO, creating combined characteristics for the TNSG and CTNSG. For TNSG, a reduced surface area of 86.4 m^2^ g^−1^ was observed with average mesopore sizes of 11.4 and 24.29 nm. The CTNSG offers a surface area of 81.3 m^2^ g^−1^ and average mesopore sizes of 11.4 and 33.86 nm. The TNP–GO composites, TNPG and CTNPG, provided surface areas of 133.4 and 139.19 m^2^ g^−1^ and average mesopore sizes of 13.2 nm and 13.1 nm. This result indicates that TNP maintains its original characteristics when composited with the GO and offers fewer areas for n–p heterojunction than the TNS. Figure 5c shows the N_2_ adsorption–desorption isotherms at 77 K and pore size distributions of the TNS, TNP, CTNSG, and CTNPG. All the selected samples exhibit Type–IV(a) isotherms according to the IUPAC classification of the adsorption isotherms [31,32]. In such cases, N_2_ condenses and evaporates irreversibly inside the adsorbent pore at high pressure, with the lower closure point of the hysteresis loops occurring at the relative pressure (P/P_0_) in the 0.7–0.8 range. This result reveals that all the adsorbents are mesoporous materials (having a pore size of 2 to 50 nm) [32] (Figure 5d, Appendix A). The adsorbents provided the H3 hysteresis loop caused by slit–shaped pores [31,32] due to the presence of stacking nanosheets (TNS–TNS), agglomerated nanoparticles (TNP–TNP) and TiO_2_–GO interfaces (TNS–GO and TNP–GO). The addition of GO and the doping of copper on the TNS and TNP alter contacts between TNS–TNS, TNP–TNP, TNS–GO and TNP–GO, leading to increased pore sizes and N_2_ condensation at higher pressure. We trust that the second and larger mesopores observed only in TNSG (24.29 nm) and CTNSG (33.86 nm) are created by the TNS–GO heterojunction that adsorbs gases and active radicals [2,31].

The optical properties of the samples were analyzed using a UV–vis spectrophotometer (UV–1800, Shimadzu). Light absorbance was monitored in a 200 to 800 nm window (Figure 6a) from a suspension of a solid sample. The suspensions were prepared by dispersing 5 mg solid power in 50 mL DI water with assistance from ultrasonication. The suspension was transferred to a 3.5 mL quartz cuvette with a 1 cm path length for the UV–vis measurement (Figure 6a). The absorbance peaks of TNP and TNPG appeared at 256 and 264 nm, responding to the UVC region of light. The TNS displayed absorbance peaks at 280 and 314 nm as the sample interacted with UVC and UVB. The TNSG and CTNSG showed excellent absorbance for UVC, UVB (317 nm) and part of the UVA (325 nm). The TNPG and CTNPG spectra peaked at 264 nm, and 280 and 327 nm, indicating photonic responses to UVC, and UVC, UVB and part of the UVA.

We obtained a light absorbance pattern and replotted the data following Tauc’s correlation [10,33] (Figure 6a (Inset)) (Equation (11)),
(11)$${\left(\alpha h\nu \right)}^{\frac{1}{n}}=A(h\nu -E_{\mathrm{bg}})$$
where A: light absorbance, α: absorption coefficient (1/dA), d: the path length of the cuvette (1 cm), hν: photon energy (eV), E_bg_: optical bandgap energy (eV), n: power factor (n = ½ for the direct transitions).

The bandgap values correlate to the amount of energy an electron needs to move across the VB to the CB and affect the number of photoelectrons involved in the photoreactions. The bandgap energy for TNS and TNP is 3.07 and 3.38 eV (Appendix A), indicating that TNS VB electrons are activated to CB electrons more effectively than the TNP. The TNSG and CTNSG reveal the bandgap energy of 2.98 and 2.66 eV, affirming improved catalyst photoactivity after compositing with GO and copper co–catalyst [34]. The bandgap for TNPG and CTNPG also reduces to 3.31 and 3.17 eV, respectively.

Photoluminescence (PL) analysis provides information regarding photoactivity, such as the charge trapping efficiency, charge immigration, and charge recombination in semiconductors [2,34]. The catalyst samples were excited at the wavelength of 320 nm, and the photonic signals emitted during the relaxation state were collected in a 350 to 600 nm window (Figure 6b). In all cases, two peaks at 415 and 440 nm represent radiated photon emission with an energy equivalent to the band gap transition of anatase TiO_2_ [34,35]. The signal from the photo emissions correlated to the electron–hole recombination rate in the photocatalyst. High emission intensity refers to a fast recombination rate, which leads to a limited number of photo–induced electrons and poor photoactivity. We compared PL signals by determining the quenching factor: a ratio of the area under the PL spectra of a photocatalyst and that of the TNS. A lower quenching factor means a lower photonic signal and a better charge separation in the photocatalyst. The quenching factor was determined to be 1.00, 1.74, 1.00, 1.56, 0.90 and 1.51 for the TNS, TNP, TNSG, TNPG, CTNSG and CTNPG, respectively. TNS is much more photoactive than TNP, indicating superior activity of the nanosheets over the nanoparticles. TNS shows the same scale of quenching factor for the TNSG and significantly reduced quenching factor for the CTNSG. The role of the Cu co–catalyst is to trap the photo–induced electrons and keep the electrons separated from holes. The co–catalyst promotes carrier separation, which helps increase the photoactivity of the CTNSG. A similar trend in quenching factor reduction was observed after compositing TNP with graphene (TNPG) and adding the copper co–catalyst (CTNPG).

### 3.2. CO_2_ Photoreduction

#### 3.2.1. Photoactivity of Photocatalyst

The prepared photocatalysts were coated on glass slides and tested in CO_2_ photoreduction (Figure 7a). The control operation was conducted in the photoreactor, filled with water–saturated CO_2_ and no photocatalyst. No photoreduction product was detected from the control operation. For the TNS, CO_2_ photoreduction yielded methane and acetone at a production rate of 4.30 and 0.42 µmol h^−1^g_cat_^−1^. The total carbon consumption is 33.3 µmol g_cat_^−1^, constituting 0.01% of the total CO_2_ inside the reactor dome. The solution used in the hydrothermal synthesis affected photoactivity of the photocatalysts. For TNS (HF + IPA), a significant decrease in CO_2_ photoreduction was observed as the production rates of 0.51 and 0.12 µmol h^−1^g_cat_^−1^were analyzed for methane and acetone with a total carbon consumption of 4.35 µmol g_cat_^−1^. For TNS (DI + HF), the photoreduction gave methane (1.85 µmol h^−1^g_cat_^−1^) and acetone (0.56 µmol h^−1^g_cat_^−1^) at a total carbon consumption of 17.65 µmol g_cat_^−1^. Effects of the post–synthesis treatment were noticed from photoactivity of the TNS–Alkaline and TNS–Thermal. CO_2_ photoreduction assisted by the TNS–Thermal photocatalyst yielded a methane production rate of 2.34 µmol h^−1^g_cat_^−1^ and an acetone production rate of 0.39 µmol h^−1^g_cat_^−1^with a total carbon consumption of 13.50 µmol g_cat_^−1^. The TNS–Alkaline is the normal TNS with fluoride ions removed, in which CO_2_ photoreduction provided methane and acetone at the production rates of 4.65 and 0.48 µmol h^−1^g_cat_^−1^ and a total carbon consumption of 30.45 µmol g_cat_^−1^. The TNP produced methane and acetone at 2.03 and 0.4 µmol h^−1^g_cat_^−1^ with total carbon consumption of 25.74 µmol g_cat_^−1^. The TNSG as a TiO_2_/graphene composite is a 2D–2D heterostructure, having TiO_2_–GO interfaces that facilitate charge separation and stabilize active radicals. The TNSG produced methane and acetone at 2.81 and 0.69 µmol h^−1^g_cat_^−1^ with total carbon consumption of 29.28 µmol g_cat_^−1^. The CTNSG improved the TNSG with a copper co–catalyst that promoted charge separation [36], providing methane and acetone at 12.09 and 0.74 µmol h^−1^g_cat_^−1^ and a total carbon consumption of 85.98 µmol g_cat_^−1^ (0.03% of the total CO_2_ inside the reactor dome). The TNPG and CTNPG yielded methane at 2.28 and 4.58 µmol h^−1^g_cat_^−1^ and acetone at 0.72 and 1.33 µmol h^−1^g_cat_^−1^, respectively. The total carbon consumption of 26.64 and 51.78 µmol g_cat_^−1^ was determined for TNPG and CTNPG. We tested the photoactivity of commercial-grade TiO_2_ nanoparticles (Degussa P25, German) in CO_2_ photoreduction. Production rates of 0.95 and 0.279 µmol h^−1^g_cat_^−1^ were determined for methane and acetone, with a total carbon consumption of 22.28 µmol g_cat_^−1^. Data on CO_2_ photoreduction experiments are collected in Appendix A.

#### 3.2.2. Effect of Humidity

Water is essential to CO_2_ photoreduction, involving hydrogenation and deoxygenation of the active radical CO_2_ (Equations (1)–(9)). We studied the effects of water in CO_2_ photoreduction, introducing water–laden CO_2_ gas at a different humidity level to the photoreactor at a total flow rate of 500 cm^3^ min^−1^. The water-saturated CO_2_ gas stream was generated by purging mass–controlled CO_2_ gas through a water liquid held inside a bubbler and mixed with a dry CO_2_ stream. Humidity is presented in percent relative humidity (%RH). It varied from 0%RH as the photoreactor was filled with dry CO_2_ gas to 20, 40, 60, 80 and 100%RH as the water vapor–saturated CO_2_ was mixed with dry CO_2_ (Figure 7b). In the case of 0%RH, the production rate for methane was 2.50 µmol h^−1^g_cat_^−1^ and acetone was 0.21 µmol h^−1^g_cat_^−1^ with a total carbon consumption of 15.65 µmol g_cat_^−1^. As %RH increased to 20 and 40%, production rates raised linearly to 4.34 and 5.12 µmol h^−1^g_cat_^−1^ for methane and 0.35 and 0.32 µmol h^−1^g_cat_^−1^ for acetone. The total carbon consumption rose to 26.95 and 30.4 µmol h^−1^g_cat_^−1^. The %RH increase to 60, 80 and 100 led to exponential growth in production rates and total carbon consumption. Methane and acetone production rates were 6.45 and 0.34 µmol h^−1^g_cat_^−1^, 9.16 and 0.42 µmol h^−1^g_cat_^−1^, and 12.09 and 0.75 µmol h^−1^g_cat_^−1^ for the 60, 80 and 100%RH. The total carbon consumptions were 37.35, 52.15 and 71.70 µmol g_cat_^−1^ for the 60, 80 and 100%RH.

Humidity in CO_2_ photoreduction is critical. After the CO_2_^−^ radical is formed through Equation (1)–(3), it either forms methane through the deoxygenation path (Equations (4), (8) and (9)) or acetone through the hydrogenation path (Equations (5)–(7)). We hypothesize that methane and acetone is produced at “the surface” and “the interface” of the photocatalyst, respectively (Figure 8). The surface site is exposed to the gas phase and is relatively scarce in water, becoming a preferential site for methane production. On the other hand, the interface contains n–p heterojunctions that stabilize intermediates and radicals, producing a more extensive product like acetone. A total reaction for CO_2_ photoreduction to acetone was proposed (Equation (12)),
3CO_2_ + 16H^+^ + 16e^−^ → CH_3_COCH_3_ + 5H_2_O(12)

Although the CO_2_ photoreduction paths need to be better understood, we trust that the formation of methane and acetone occurs parallelly and can only be competitive at consuming CO_2_^−^ and other active radicals. In our previous work [12], we conducted CO_2_ photoreduction in an aqueous system using a copper–doped dititanate/graphene heterostructure (CTG) photocatalyst: the other 2D–2D heterostructure. The reactions yielded methanol, ethanol, IPA, and acetone, with no detectable amount of methane gas. The CTG was tested in the gas phase photoreduction, producing only methane and acetone at 4.99 and 0.43 µmol h^−1^g_cat_^−1^with a total carbon consumption of 31.5 µmol g_cat_^−1^ (100%RH).

#### 3.2.3. Effect of the Light Source

The effect of a light source on CO_2_ photoreduction was investigated using three available light sources (Figure 7c): a 16 Watts mercury lamp (Philips, ML TUV 16 W, λ ~240 nm) and a 160 Watts mercury lamp (Philips, ML 160 W E27, λ ~360, 440 and 540 nm), and a 250 Watts mercury lamp (Master HPI Plus 250 W, λ ~460, 540 and 600 nm). The CTNSG absorbs part of visible light and responds to UVC, UVB and UVA, as observed from the UV-vis spectra. For the 16 Watts mercury source, the illuminated CTNSG produced 6.84, 1.23 and 3.07 µmol h^−1^g_cat_^−1^ for methane, acetone, and methanol with a total carbon consumption of 86.4 µmol g_cat_^−1^. The 160 Watts mercury lamp yielded 12.09, 0.75 and 0.00 µmol h^−1^g_cat_^−1^ for methane, acetone, and methanol with a total carbon consumption of 71.7 µmol g_cat_^−1^. For the 250 Watts mercury lamp, the CTNSG produced only methane and acetone at a production rate of 14.09 and 0.56 µmol h^−1^g_cat_^−1^ and a total carbon consumption of 78.9 µmol g_cat_^−1^. Illumination from a combination of the 16 Watts and 160 Watts light sources assist CTNSG in generating methane, methanol, and acetone at 6.35, 1.16 and 2.91 µmol h^−1^g_cat_^−1^at a total carbon consumption of 81.2 µmol g_cat_^−1^. The photoreduction quantum efficiency (PQE) was defined as follows [37]:(13)$$\mathrm{PQE}\;(\%)\;=\;\frac{{n\;*\;\mathrm{production}\;\mathrm{rate}\;(\mu \mathrm{mol}\;s}^{-1})}{{\;\mathrm{Incident}\;\mathrm{photon}\;\mathrm{rate}\;(\mu \mathrm{mol}\;s}^{-1})}\text{×}\;100\;$$
in which “n” is the number of photoelectrons needed to form a product, and the incident photon rate is estimated using Equation (14) [37,38].
(14)$$\mathrm{Incident}\;\mathrm{photon}\;\mathrm{rate}\;(\text{µ}\mathrm{mol}\;s{-}^{1})=\frac{\mathrm{Light}\;\mathrm{intensity}\;\left({W\;\mathrm{cm}}^{-2}\right)\;*\;\mathrm{Wave}\;\mathrm{length}\;\mathrm{of}\;\mathrm{light}\;\left(m\right){*\;A\;(\mathrm{cm}}^{2})}{\;\mathrm{Planck}\text{’}s\;\mathrm{constant}\;\left({J\;s\;\mathrm{mol}}^{-1}\right)*\;\mathrm{Photon}\;\mathrm{density}\;\left({m\;s}^{-1}\right)*\;\mathrm{Avogadro}\text{’}s\;(\mathrm{mol})}$$
where light intensity is the energy per second per unit area, the wavelength is the wavelength of light, and A is the projected area of the reactor (78.5 cm^2^). Planck’s constant, photon density and Avogadro’s number are 6.63 × 10^−34^ J s mol^−1^, 3 × 10^8^ m s^−1^, and 6.023 × 10^23^ molecules mol^−1^, respectively. The PQE is an indicator of photoactivity, providing quantitative data on how effectively a photocatalyst utilizes photonic energy from light (Figure 7b (inset)). We determined the PQE for methane, acetone, and methanol and combined them to obtain the total PQE in percent (%). When illuminated by the 16 Watts mercury lamp, CTNSG exhibited PQE of 23.19 × 10^−2^, 20.82 × 10^−2^ and 3.13 × 10^−2^ for methane, acetone, and methanol with a total PQE of 47.15 × 10^−2^%. As we switched the light source to the 160 Watts lamp, the CTNSG provided PQE of 0.57 × 10^−2^, 0.00 and 0.07 × 10^−2^ for methane, acetone, and methanol with a total PQE of 0.64 × 10^−2^%. With the higher light intensity of the 250 Watts lamp, CTNSG yielded a lower PQE of 0.36 × 10^−2^, 0.00 and 0.03 × 10^−2^% for methane, acetone, and methanol and a total PQE of 0.39 × 10^−2^%. Illuminating CTNSG with a combination of 16 Watts and 160 Watts lamps gave methane, acetone, and methanol at the PQE of 21.83 × 10^−2^, 20.01 × 10^−2^, and 2.99 × 10^−2^ with a total PQE of 44.84 × 10^−2^%.

The effect of the light sources on CO_2_ photoreduction follows concepts of the photoelectric effect, in which the photon’s energy is inversely proportional to the wavelength of the electromagnetic wave. Photons from a shorter wavelength provide photo–induced electrons with a more intense plasmonic energy, forming a larger product molecule. The amount of product correlates to the number of photons bombarded and absorbed on a photocatalyst and corresponds to the power of the light source. The 16 Watts mercury lamp supplies photoelectrons with high plasmonic energy (UVC), promoting the production of methanol at the surface (competing with methane production) and acetone at the interface of the CTNSG. The 160 Watts lamp gives more photons for the photocatalyst illumination but larger wavelengths in UVA and visible region. The photons yield less plasmonic energy to the CTNSG, producing more methane (small product) and less acetone (massive product). The 250 Watts lamp offers even more photons for illumination but with less plasmonic energy (visible region), resulting in more methane and fewer acetone productions. A combination of 16 Watts and 160 Watts light sources emits UVC, UVA and visible light, activating CTNSG to create methane, methanol, and acetone. The combined light source provides the same scale of production rate for methane, methanol, and acetone as that of the 16 Watts mercury lamp. This result indicates that methane and methanol formations are competing and may occur at the surface of the TNS.

We benchmarked our results with other research groups (Table 1), relying on the CTNSG photocatalyst with 160 Watts or 16 Watts mercury lamps. To our knowledge, we are the first to report acetone vapor as a product of gas–phase CO_2_ photoreduction. Alkanad and his team [23] synthesized anatase TiO_2_ nanotubes using a one–step hydrothermal technique and utilized them for CO_2_ photoreduction. The nanotubes were illuminated by a 300 Watts Xenon lamp (Xe), yielding only methane at an unprecedented production rate of 47 µmol g_cat_^−1^h^−1^. Rodriguez and his team [39] applied a pressurized supercritical CO_2_–ethanol medium to synthesize copper–TiO_2_/rGO composites. The composite facilitated the CO_2_ photoreduction (influenced by 450 Watts Xe lamp) and produced methane and carbon monoxide at 2.20 and 5.20 µmol g_cat_^−1^h^−1^), respectively. The group of Larimi [11] demonstrated electrostatically induced chemical deposition of platinum (Pt) and TiO_2_ on rGO, multi–walled carbon nanotubes (MWCNT) and single–walled carbon nanotubes (SWCNT). Under the influence of 15 Watts light bulb, the Pt–TiO_2_/rGO, Pt–TiO_2_/MWCNT and Pt–TiO_2_/SWCNT yield methane as the only product from CO_2_ conversion at production rates of 0.50, 1.90 and 0.70 µmol g_cat_^−1^h^−1^, respectively. The group of Zhao [40] fabricated the core–shell structure of Pt–TiO_2_/rGO following hydrothermal technique and chemical reduction. The Pt–TiO_2_/rGO was illuminated by a 300 Watts Xe lamp, converting CO_2_ to methane at a production rate of 4.70 µmol g_cat_^−1^h^−1^.

#### 3.2.4. Reusability and Stability of Photocatalyst

Recycling catalysts is crucial in determining the practical use and development of heterogeneous photocatalysts. During the first round of CO_2_ exposure, CTNSG exhibited a photocatalytic efficiency of 100%, giving methane and acetone production rates of 11.7 and 0.3 µmol h^−1^g_cat_^−1^ and a total carbon consumption of 67.4 µmol g^−1^h^−1^ (Figure 7d). After the first exposure, the photoreactor was flooded with a fresh water–saturated CO_2_ stream for 10 min before the CO_2_ photoreduction. The second exposure of the CTNSG to the CO_2_ yielded a significant drop of 30% in total carbon consumption and 30 and 6% in methane (8.4 µmol h^−1^g_cat_^−1^) and acetone (0.3 µmol h^−1^g_cat_^−1^) production rates. The trend of decreased photoactivity continued for the third, fourth and fifth exposures as the total carbon consumption dropped to 61.1, 51.8, and 32.1%, respectively.

We investigated the matter and analyzed the CTNSG with FT-IR after the fifth round of CO_2_ photoreduction (Appendix A). For the used CTNSG, IR transmittance peaks at 3320, 1630, 580 and 470 cm^−1^ were assigned to −OH, C=C, Ti–O–C and Ti–O, respectively. The IR peak, representing the −OH group, shifts slightly to the right compared to the fresh CTNSG, indicating a decrease in the photocatalyst’s mass. The signals from C=C in the new and used CTNSG occur at the exact wavenumber (1630 cm^−1^) with relatively equal intensity. The result indicates no significant change in the amount of unsaturated alkene (C=C), which only exists in the GO structure. The IR signals for Ti–O–C and Ti–O are significantly lower for the used CTNSG, which suggests a reduction in CTNS–GO interactions [41] and the amount of CTNS on GO. The loss in CNTS–GO interactions is the primary cause of photoactivity deterioration since it affects the charge transfer and charge separation ability of the composites. The XRD patterns (Appendix A) reveal similar anatase structures of TNS on the fresh and used CTNSG with crystallite sizes of 14.0 and 11.8 nm, respectively. The patterns provide an informative analysis that the crystal structure and crystallite size are not affected by assisting in CO_2_ photoreduction. The physical geometry of the used CTNSG, as analyzed by SEM (Appendix A), exhibits that CTNS is 21.83 ± 0.7 nm in size and 6.2 ± 0.1 nm in thickness. The CTNS dimensions on the used CTNSG are on the same scale as those of the fresh CTNSG, indicating no physical damage to the composite after CO_2_ photoreduction.

## 4. Conclusions

The CTNSG 2D–2D heterostructure was synthesized using a one–step hydrothermal technique with assistance from the HF soft template. The CTNSG was applied as a photocatalyst in gas–phase CO_2_ photoreduction and demonstrated superior photoactivity, producing mainly methane and acetone. The excellent photoactivity was attributed to the n–p heterojunctions in the heterostructures, which house charged radicals and reduce the electron-hole recombination rate of the CTNSG. The CTNSG-GO heterojunctions provided suitable pore sizes for CO_2_ adsorption. The humidity level and type of light sources affect the production rate and products from CO_2_ photoreduction. The reaction mechanisms are proposed here as we trust that the CTNS surfaces and CTNSG–GO interfaces are responsible for forming methane and acetone, respectively. The approach of gas–phase CO_2_ photoreduction at ambient conditions can contribute to reducing CO_2_ emissions from industrial sectors or be utilized as a part of a direct air capture unit.

## Figures and Tables

**Figure 1 nanomaterials-13-00320-f001:**
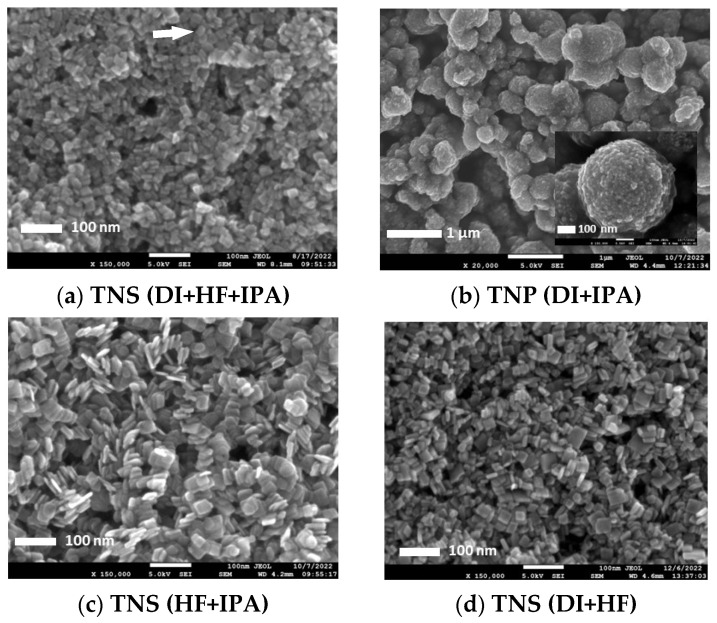
SEM images showing physical geometry of TNS, TNP and composite samples.

**Figure 2 nanomaterials-13-00320-f002:**
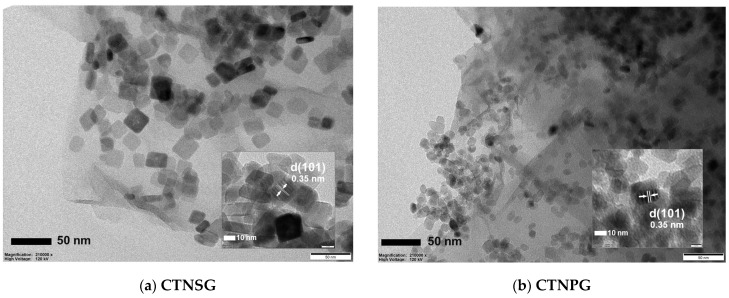
TEM images of: (**a**) CTNSG; and (**b**) CTNPG.

**Figure 3 nanomaterials-13-00320-f003:**
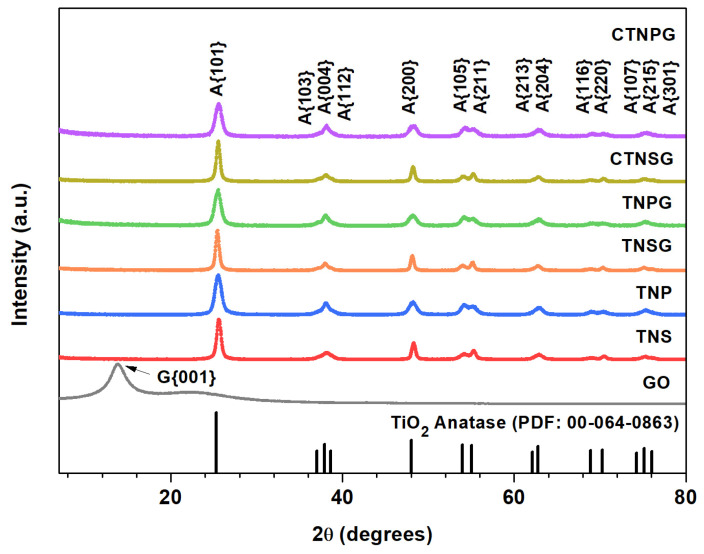
The XRD patterns of GO, TNS, TNP TNSG, TNPG, CTNSG and CTNPG.

**Figure 4 nanomaterials-13-00320-f004:**
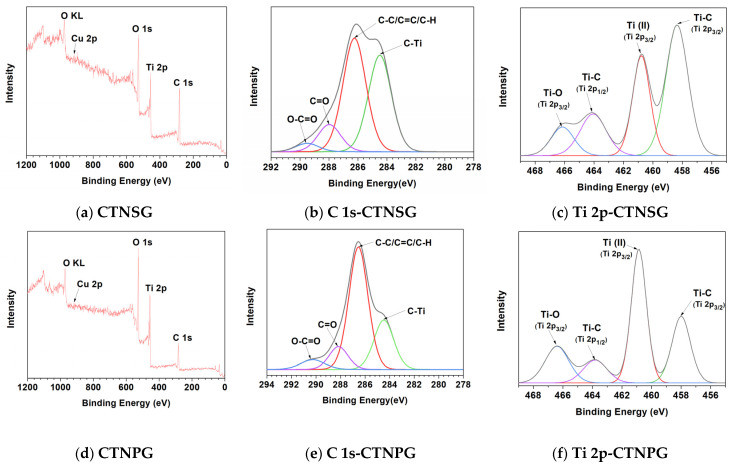
The XPS results of a wide scan, narrow scan of Ti 2p, and narrow scan of C 1s of CTNSG (**a**–**c**) and CTNPG (**d**–**f**).

**Figure 5 nanomaterials-13-00320-f005:**
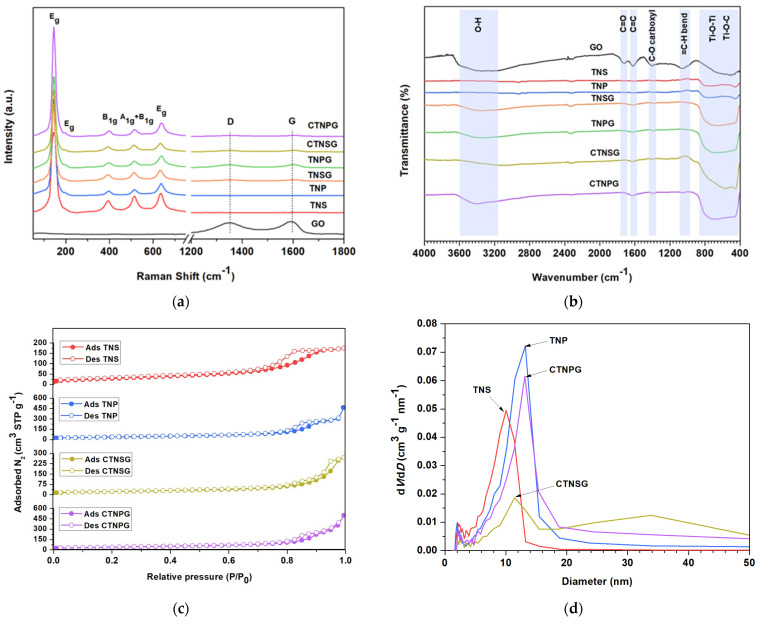
Raman spectra (**a**); and FTIR spectra (**b**); of the TNS, TNP, TNSG, TNPG, CTNSG and CTNPG, and the N_2_ adsorption–desorption isotherms (**c**); and the pore−size distribution curves (**d**) of the TNS, TNP, CTNSG and CTNPG.

**Figure 6 nanomaterials-13-00320-f006:**
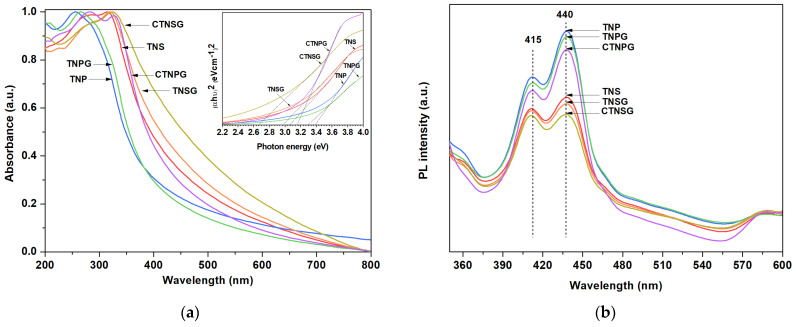
(**a**) UV−vis spectra and Tauc’s plot (Inset) of the TNS, TNP, TNSG, TNPG, CTNSG, and CTNPG; and (**b**) Photoluminescence spectra of TNS, TNP, TNSG, TNPG, CTNSG and CTNPG.

**Figure 7 nanomaterials-13-00320-f007:**
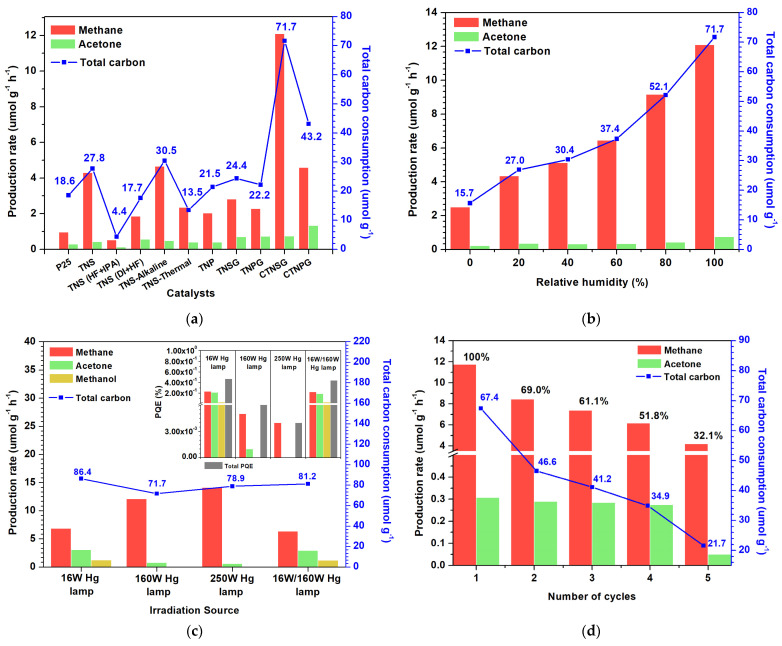
Production rates of methane and acetone (left axis) and total carbon consumption rate (right axis) from the CO_2_ photoreduction, corresponded to various photocatalysts (**a**); effect of humidity on CO_2_ photoreduction (CTNSG as the photocatalyst) (**b**); effect of light source and (inset) photo quantum efficiency (**c**); reusability of the CTNSG (**d**).

**Figure 8 nanomaterials-13-00320-f008:**
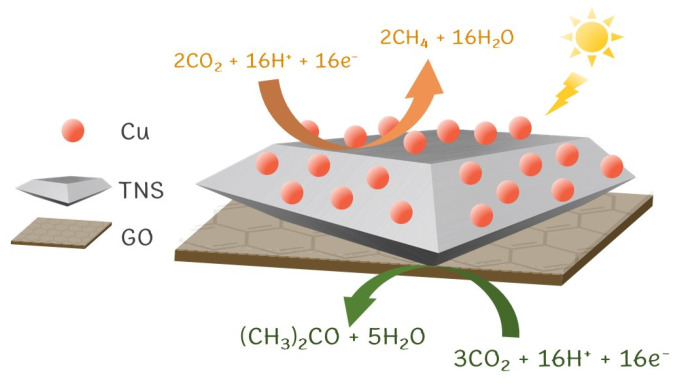
Schematic diagram proposing reaction mechanisms for the photoreduction of CO_2_ on CTNSG.

**Table 1 nanomaterials-13-00320-t001:** Comparison of gas-phase CO_2_ photoreduction products obtained from a variety of TiO_2_−based photocatalyst.

Catalysts	Condition	Production Rate (µmol h^−1^g_cat_^−1^)			Reference
		Methane	Methanol	Acetone	
Copper−TiO_2_ nanosheet/GO (CTNSG)	-10 mg_cat_ -160 Watts mercury -5 h reaction time	12.09	0	0.74	This work
Copper−TiO_2_ nanosheet/GO (CTNSG)	-10 mg_cat_ -16 Watts mercury -5 h reaction time	6.84	1.23	3.07	This work
TiO_2_ nanotubes	-50 mg_cat_ -300 Watts Xe -5 h reaction time	47.00	-	-	[23]
Copper−TiO_2_/rGO	-50 mg_cat_ -450 Watts Xe -3 h reaction time	2.20	0	0	[39]
Pt−TiO_2_/rGO Pt−TiO_2_/MWCNT Pt−TiO_2_/SWCNT	-100 mg_cat_ -15 Watts energy-saving daylight lamp (Philips) -12 h reaction time	0.50 1.90 0.70	0	0	[11]
Pt−TiO_2_ nanosheet/rGO	-300 Watts Xe -8 h reaction time	4.70	0	0	[40]

## Data Availability

Not applicable.

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
