# Peer review of "The Photocatalytic Conversion of Carbon Dioxide to Fuels Using Titanium Dioxide Nanosheets/Graphene Oxide Heterostructure as Photocatalyst"

_nanomaterials, 2023, doi:10.3390/nano13020320_

Round 1

Reviewer 1 Report

In this paper, the authors have proposed the fabrication and characterization of copper-doped TiO2 nanosheets/graphene oxide composite as a photocatalyst using a one-step hydrothermal technique for the molecular transformation of carbon dioxide to hydrocarbon and alcohol-type chemicals. It is an interesting topic for the fabrication and characterization of a heterostructure, the copper-doped TiO2 nanosheets and graphene oxide, for the promotion of electron-hole separation in the photocatalyst. However, the discussions for this version to support the big achievement of these fields are weak. Therefore, the Authors need the revision the manuscript for publication in “nanomaterials” journal. Some questions and suggestions are as followed;

[1] We suggest that authors should discuss how the feed content of each component was determined in fabricating the nanocomposites. We believe that the photocatalytic conversion of this system can be affected by feed content in the nanocomposites.

[2] We suggest that it would be better to include the detailed results in the Conclusions section.

[3] The form of references described in the References part does not match the guideline of the “nanomaterials” journal. The authors should revise the references’ form accurately.

Reviewer 2 Report

The authors reported that the TiO2 nanosheets were composited with graphene oxide and doped with copper oxide in the hydrothermal process to create the copper-TiO2 nanosheets/graphene oxide (CTNSG). The CTNSG exhibited outstanding photoactivity in converting CO2 gas to methane and acetone. This article is clear in thought and detailed in content. I suggest it be accepted after minor revisions.

1. Please provide the standard XRD patterns and PDF card numbers of GO and TiO2.

2. In Figure 6a (UV-vis spectra and Tauc’s plot (Inset)), please confirm the correctness of the Tauc plots. You can refer to the equation Ahν = α(h – Eg)n/2 from “Applied Catalysis B: Environmental 2017, 217, 378–387” and “Nanoscale, 2019, 11, 10439”.

Reviewer 3 Report

The manuscript discusses CO2 photoreduction by TiO2/GO catalyst. The authors should address the main concerns before considering being published in Nanomaterials.

1. In part 3.2.3, the authors discuss the effect of the light source. To me, it is hard to compare the three completely different light sources because of the different light wavelengths, even though all are called mercury lamps. Formula 14 can only be used if the ratio of the light wavelength is notified.

It would be easier to understand if the author could find two or three lamps with the same power but with different mono wavelengths. An important paper using 380 nm UV light compared with UV lamps should be cited here (10.1021/acsnano.0c09155).

2. The format of the references should be double-checked. The first letter of the journal names should be uppercase, for example, RSA Advance (A uppercase), ACS Omega (O uppercase), etc..
